# Towards Deeper Deep Reinforcement Learning with Spectral Normalization

**Johan Bjorck**[*],  **Carla P. Gomes,**  **Kilian Q. Weinberger**
Cornell University

## Abstract

In computer vision and natural language processing, innovations in model architecture that increase model capacity have reliably translated into gains in performance. In stark contrast with this trend, state-of-the-art reinforcement learning (RL) algorithms often use small MLPs, and gains in performance typically originate from algorithmic innovations. It is natural to hypothesize that small datasets in RL necessitate simple models to avoid overfitting; however, this hypothesis is untested. In this paper we investigate how RL agents are affected by exchanging the small MLPs with larger modern networks with skip connections and normalization, focusing specifically on actor-critic algorithms. We empirically verify that naïvely adopting such architectures leads to instabilities and poor performance, likely contributing to the popularity of simple models in practice. However, we show that dataset size is not the limiting factor, and instead argue that instability from taking gradients through the critic is the culprit. We demonstrate that spectral normalization (SN) can mitigate this issue and enable stable training with large modern architectures. After smoothing with SN, larger models yield significant performance improvements — suggesting that more "easy" gains may be had by focusing on model architectures in addition to algorithmic innovations.

## 1  Introduction

In computer vision and natural language processing (NLP), competitive models are growing increasingly large, and researchers now train billion-parameter models [13, 36]. The earliest neural networks were often shallow [40] with performance dropping for excessively deep models [29]. However, ever since the introduction of batch normalization [31] and residual connections [29], performance has improved more or less monotonically with model scale [64]. As a result, competitive models in computer vision and NLP are growing ever larger [10, 56], and further architectural innovations are continuously researched [14].

In stark contrast with this trend, state-of-the-art (SOTA) reinforcement learning (RL) agents often rely on small feedforward networks [35, 37, 39] and performance gains typically originate from algorithmic innovations such as novel loss functions [58, 62, 68] rather than increasing model capacity. Indeed, a recent large-scale study has shown that large networks can harm performance in RL [2]. It is natural to suspect that high-capacity models might overfit in the low-sample regime common in RL evaluation. Imagenet contains over a million unique images whereas RL is often evaluated in contexts with fewer environment samples [37, 39]. However, to date, this overfitting hypothesis remains largely untested.

To address this, we study the effects of using larger *modern* architectures in RL. By modern architectures we mean networks with high capacity, facilitated by normalization layers [6, 31] and skip connections [29]. We thus depart from the trend in RL of treating networks as black-box function

---

[*]Correspondence to: Johan Bjorck <njb225@cornell.edu>

35th Conference on Neural Information Processing Systems (NeurIPS 2021).

approximators. To limit the scope and compute requirements we focus on continuous control from pixels [65] with two actor-critic agents based on the Soft Actor-Critic (SAC) [25, 35] and Deep Deterministic Policy Gradients (DDPG) [43, 71] algorithms. Actor-critic methods form the basis of many SOTA algorithms for continuous control [35, 38, 39]. As can be expected, we demonstrate that naïvely adopting modern architectures leads to poor performance, supporting the idea that RL does not benefit from larger models. However, we show that the issue is not necessarily overfitting, but instead, that training becomes unstable with deeper modern networks. We hypothesize that taking the gradient of the actor through the critic network creates exploding gradients [53] for deeper networks. We connect this setup with generative adversarial networks (GANs) [22], and propose to use a simple smoothing technique from the GAN literature to stabilize training: *spectral normalization* [47].

We demonstrate that this simple strategy allows the training of larger modern networks in RL without instability. With these fixes, we can improve upon state-of-the-art RL agents on competitive continuous control benchmarks, demonstrating that improvements in network architecture can dramatically affect performance in RL. We also provide performance experiments showing that such scaling can be relatively cheap in terms of memory and compute time in RL from pixels. Our work suggests that model scaling is complementary to algorithmic innovations and that this simple strategy should not be overlooked. We summarize our contributions as follows:

- We verify empirically that large modern networks fail for two competitive actor-critic agents. We demonstrate dramatic instabilities during training, which casts doubt on overfitting being responsible.

- We argue that taking the gradients through the critic is the cause of this instability. To combat this problem we propose to adopt spectral normalization [47] from the GAN literature.

- We demonstrate that this simple smoothing method enables the use of large modern networks and leads to significant improvements for SOTA methods on hard continuous control tasks. We further provide evidence that this strategy is computationally cheap in RL from pixels.

## 2 Background

### 2.1 Reinforcement Learning

Reinforcement learning tasks are often formulated as Markov decision processes (MDPs), which can be defined by a tuple $(\mathcal{S}, \mathcal{A}, P, r)$ [63]. For continuous control tasks the action space $\mathcal{A}$ and the state space $\mathcal{S}$ are continuous and can also be bounded. Each dimension of the action space might for example correspond to one joint of a humanoid walker. At time step $t$, the agent is in a state $\mathbf{s}_t \in \mathcal{S}$ and takes an action $\mathbf{a} \in \mathcal{A}$ to arrive at a new state $\mathbf{s}_{t+1} \in \mathcal{S}$. The transition between states given an action is random with transition probability $P : \mathcal{S} \times \mathcal{S} \times \mathcal{A} \to [0, \infty)$. The agent receives a reward $r_t$ from the reward distribution $r$ at each timestep $t$. The goal is typically to find a policy $\pi : \mathcal{S} \to \mathcal{A}$ that maximizes expected discounted reward $\mathbb{E}[\sum_t \gamma^t r_t]$, where $0 < \gamma < 1$ is a pre-determined constant. In practice, it is common to measure performance by the cumulative rewards $\sum_t r_t$.

### 2.2 Actor-Critic Agents

Actor-critic methods have roots in Q-learning [69] and form the backbone of many recent state-of-the-art algorithms [35, 38, 39]. We focus on two popular and performant actor critic methods: SAC [25] and DDPG [43]. Given some features $\phi(\mathbf{s})$ that can be obtained from any state $\mathbf{s} \in \mathcal{S}$, the actor-network maps each state-feature $\phi(\mathbf{s})$ to a distribution $\pi_\theta(\mathbf{s})$ over actions. The action distribution is known as the *policy*. For each dimension, the actor-network outputs an independent normal distribution, where the mean $\boldsymbol{\mu}_\theta(\mathbf{s})$ and possibly the standard deviation $\boldsymbol{\sigma}_\theta(\mathbf{s})$ come from the actor-network. The action distribution is then obtained as

$$\pi_\theta(\mathbf{s}) = \tanh(\boldsymbol{\mu}_\theta(\mathbf{s}) + \boldsymbol{\epsilon} \odot \boldsymbol{\sigma}_\theta(\mathbf{s})), \qquad \boldsymbol{\epsilon} \sim \mathcal{N}(0, \mathbf{1}). \tag{1}$$

The $\tanh$ non-linearity is applied elementwise, which bounds the action space to $[-1, 1]^n$ for $n = dim(\mathcal{A})$. Sampling $\boldsymbol{\epsilon}$ allows us to sample actions $\mathbf{a}_\theta$ from the policy $\pi_\theta(\mathbf{s})$. DDPG typically adds the exploration noise $\boldsymbol{\epsilon}$ outside the $\tanh$ non-linearity. To promote exploration, SAC instead adds the entropy of the policy distribution $H(\pi_\theta(\mathbf{s}_t))$ times a parameter $\alpha$ to the rewards. The *critic* network outputs a Q-value $Q_\psi(\mathbf{a}, \mathbf{s})$ for each state $\mathbf{s}$ and action $\mathbf{a}$. In practice, one can simply

concatenate the feature $\phi(\mathbf{s})$ with the action $\mathbf{a}$ and feed the result to the critic. The critic network is trained to minimize the soft Bellman residual:

$$\min_{\psi} \mathbb{E}\left(Q_{\psi}(\mathbf{a},\mathbf{s}_t) - \left[r_t + \gamma\mathbb{E}[\hat{Q}(\mathbf{a}_{t+1},\mathbf{s}_{t+1}) + \alpha H(\pi)]\right]\right)^2 \tag{2}$$

Here, $r_t$ is the obtained reward and $\mathbb{E}[\hat{Q}(\mathbf{a}_{t+1},\mathbf{s}_{t+1})]$ is the Q-value estimated by the *target* critic – a network whose weights are the exponentially averaged weights of the critic. One can also use multi-step targets for the rewards [62]. The loss (2) is computed by sampling transitions from a replay buffer [48]. Since the Q-values that the critic outputs measure expected discounted rewards, the actor should simply take actions that maximize the output of the critic network. Thus, to obtain a gradient step for the actor, both SAC and DDPG sample an action $\mathbf{a}_{\theta}$ from $\pi_{\theta}(\mathbf{s}_t)$ and then take derivatives of $Q_{\psi}(\mathbf{a}_{\theta},\mathbf{s})$ with respect to $\theta$. That is

$$\nabla_{\theta} = \frac{\partial Q_{\psi}(\mathbf{a}_{\theta},\mathbf{s})}{\partial\theta} \tag{3}$$

As the critic is a differentiable neural network, the derivatives are taken *through* the Q-values $Q_{\psi}(\mathbf{a}_{\theta},\mathbf{s})$. In practice, the features $\phi(\mathbf{s})$ can be obtained from a convolutional network trained by backpropagating from (2). For further details, see Haarnoja et al. [25], Lillicrap et al. [43].

## 3 Motivating Experiments

### 3.1 Experimental Setup

For experimental evaluation, we focus on continuous control from pixels, a popular setting relevant to real-world applications [33, 34]. Specifically we evaluate on the DeepMind control suite [65], which has been used in [26, 27, 35, 39]. We use the 15 tasks considered in Kostrikov et al. [35] and evaluate after 500,000 samples, which has become a common benchmark [35, 39, 41]. If learning crashes before 500,000 steps (this occasionally happens for modern networks), we report the performance before the crash. We will primarily focus on the image augmentation based SAC agent of Kostrikov et al. [35], known as DRQ, and adopt their hyperparameters (listed in Appendix B). This agent reaches state-of-the-art performance without any nonstandard bells-and-whistles, and has an excellent open-source codebase. Unless specifically mentioned, all figures and tables refer to this agent. Its network consists of a common convolutional encoder that processes the image into a fixed-length feature vector. Such feature vectors are then processed by the critic and actor-network, which are just feedforward networks. We will refer to these feedforward networks as the *heads*. The convolutional encoder consists of four convolutional layers followed by a linear layer and layer normalization [6] and the heads consist of MLPs with two hidden layers that concatenate the actions and image features; see Appendix C for details. In Table 1 we also consider the augmentation-based DDPG agent of Yarats et al. [71], using the default hyperparameters proposed therein. Both agents use the same network design. In all experiments, we only vary the head architecture. We evaluation over ten seeds.

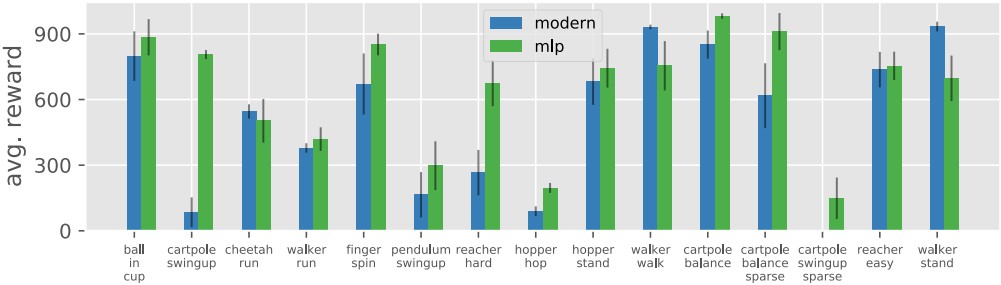

Figure 1: Performance when using a modern network architecture with skip connections [29] and normalization [6], averaged across ten seeds. Using modern architectures does not enable deeper networks; instead, performance often decreases — sometimes catastrophically — while increasing for a handful of tasks. This is in stark contrast with supervised learning where performance typically improves monotonically with model capacity.

## 3.2 Testing Modern networks

Two crucial architectural innovations which have enabled deep supervised models are residual connections [29] and normalization [31]. We test if adopting such modern techniques allows us to successfully scale networks. Specifically, we use the feedforward network (and not the attention module) found in a Transformer block [67] and will refer to this architecture as *modern*. Each block consists of two linear transformations $w_1, w_2$ with a ReLu between and layer normalization [6] that is added to a residual branch. I.e. the output of a layer is $x + w_2\text{Relu}(w_1\text{norm}(x))$. Compared to the original architecture, normalization is applied before the feedforward blocks instead of after, which is now favored in practice [50, 70]. We use four transformer blocks for the critic and two for the actor. Results from these experiments are shown in Figure 1, with one standard error given. We see that these modifications often decrease performance. Thus, actor-critic agents have seemingly not reached the level where model capacity improves performance monotonically, and tricks used in supervised learning do not seem to suffice. This is perhaps not very surprising as state-of-the-art RL agents typically use simple feedforward networks.

## 3.3 Testing Deeper Networks

Whereas naively adopting modern networks failed, we now investigate if simply making the MLP networks deeper is a competitive strategy. To do this, we increase the number of MLP layers in the actor and critic — from two to four hidden layers — while keeping the width constant. Results for the two configurations are shown in Figure 2, with one standard error given. We see that for some tasks, the performance improves, but for many others, it decreases. Simply scaling the network does not monotonically improve performance, as typically is the case in supervised learning. For practitioners, the extra computational burden might not make it worthwhile to increase network scale for such dubious gains.

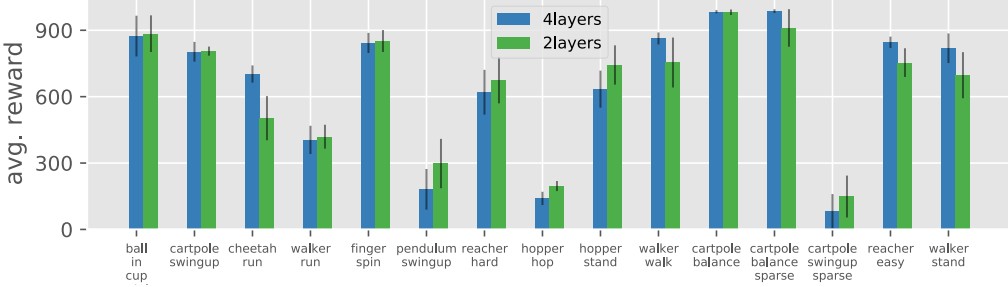

Figure 2: The performance when making the networks deeper. Performance is averaged across ten seeds and plotted for 15 tasks from the DeepMind control suite [65]. Making the network deeper does not monotonically improve performance as is typically the case in supervised learning. Instead, some tasks benefit, and some tasks suffer. As we shall see later, it is possible to improve the performance by stabilizing training, suggesting that deeper networks cause some instability.

## 3.4 Is it overfitting?

Both making the networks deeper and adopting modern methods such as normalization and residual connections do not monotonically improve performance, and can occasionally lead to catastrophic drops in performance. It is natural to think that the small number of environment transitions available leads to overfitting for large models, which is known to be harmful in RL [76]. Indeed, since training and collecting samples are interleaved in RL, RL agents need to generate useful behavior when only 10 % of the training has elapsed, at which point only 10 % of the sample budget is available. Such an overfitting hypothesis is relatively easy to probe. As per eq. (2), the critic is trained to minimize the soft Bellman residual. If overfitting truly was the issue, we would expect the loss to decrease when increasing the network capacity. We investigate this by comparing a modern network against a smaller MLP in an environment where the former performs poorly — cartpole-swing-up. First, we study the critic loss as measured over the replay buffer — which plays the role of training loss in supervised learning. Results averaged over five runs are shown in Figure 3. The losses initially start on the same scale, but the modern network later has losses that increase dramatically — suggesting

that overfitting is not the issue. Instead, training stability could be the culprit, as stable training should enable the network to reach lower training loss. To probe this, we simply plot the norm of the gradients for the critic and actor networks and see that they indeed are larger for the large modern networks. Furthermore, the gradient magnitude increases during training and shows large spikes, especially for the actor. This suggests that training stability, rather than overfitting, is the issue.

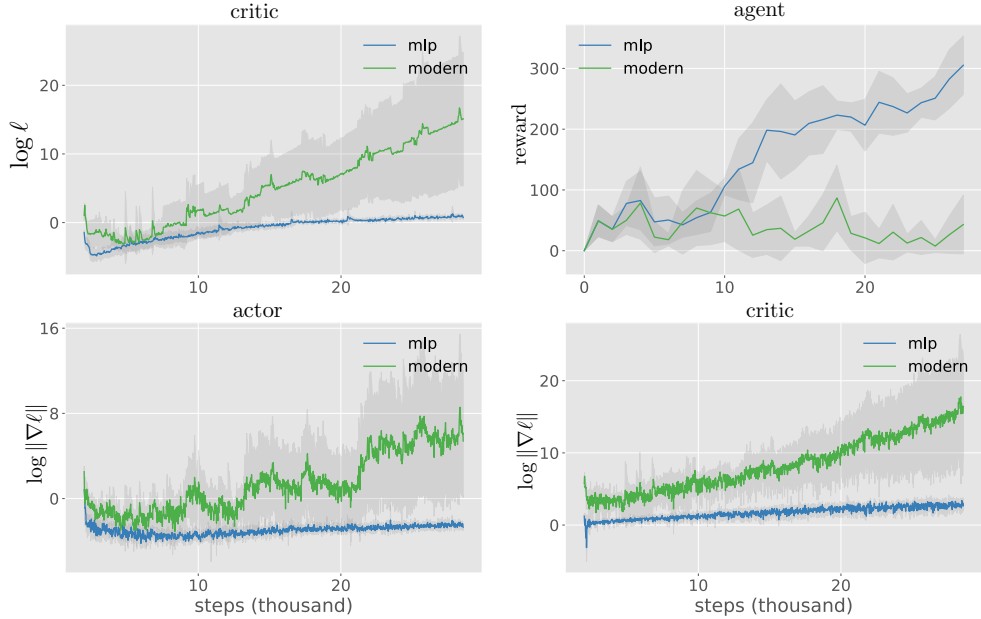

Figure 3: Training dynamics when using a small MLP or a deeper network with normalization and residual connections. **Top left.** The loss, as per eq. (2), of the critic during training. The loss does not decrease when making the network larger, suggesting that overfitting is not the issue. **Top right.** The rewards obtained. The deeper network fails to learn and improve its performance. **Bottom left & right.** The gradient of the actor and critic during training. For the deeper network, training becomes unstable with occasional spikes and large and growing gradients. Notice the logarithmic scale.

## 4  Stabilizing with Spectral Normalization

To improve the stability of actor-critic algorithms, let us consider a simple model. We will assume that the action space has one degree of freedom, i.e. $\mathcal{A} = \mathbb{R}$. We also assume that we have some fixed features $\phi$ which maps states to $n$-dimensional vectors, i.e. $\phi(\mathcal{S}) = \mathbb{R}^n$. The critic and actor are both modeled as MLPs with $N$ layers and a nonlinearity $f$ (e.g. ReLu) applied elementwise after each layer. The weights are given as matrices $w_i^a$ or $w_i^c$ for layer $i$ in the actor and critic respectively. We then have $y_i = f(w_i x_i)$ for layer $i$. If we let the functions representing the actor and critic be denoted by $A$ and $Q$, and let $\|$ denote vector concatenation, we have:

$$A(\mathbf{s}) = \Big( \prod_i f \circ w_i^a \Big) \circ \phi(\mathbf{s}) \qquad Q(\mathbf{s}) = Q(A(\mathbf{s}), \phi(\mathbf{s})) = \Big( \prod_i f \circ w_i^c \Big) \circ \big( A(\mathbf{s}) \| \phi(\mathbf{s}) \big) \qquad (4)$$

Recall that the Lipschitz constant of a function $f$ is the smallest constant $C$ such that $\|f(x) - f(y)\| \leq C\|x - y\|$ for all $x, y$. It is straightforward to bound the Lipshitz constant of the critic, as it is made up of transformations with bounded constants themselves. The Lipschitz constant of the linear layer $w_i$ is the operator norm $\|w_i\|$, equal to the largest singular value $\sigma_{\max}$ of $w_i$. We have:

$$\|Q(\mathbf{s})\| \leq \big( \|A(\mathbf{s})\| + \|\phi(\mathbf{s})\| \big) \prod_i \|f\| \|w_i^c\| \qquad (5)$$

Here $\|f\|$ is the Lipschitz constant of the function $f$, which for Relu is 1. Equation (5) bounds the smoothness of the critic in the forward pass. A function that is $L$-Lipschitz smooth also has a gradient bounded by $L$. Thus, if we could ensure that the critic is $L$-smooth, that would also imply

that gradients $\frac{\partial Q_\psi}{\partial \mathbf{a}_\theta}$ being propagated into the actor, as per (3), are bounded. Equation (5) suggest that the critic could be made smooth if the spectral norms of all layers are bounded. Fortunately, there is a method from the GAN literature which achieves this: spectral normalization [47]. Spectral normalization divides the weight $W$ for each layer by its largest singular value $\sigma_{\max}$ which ensures that all layers have operator norm 1. The singular value can be expensive to compute in general, and to this end, spectral normalization uses two vectors $u$ and $v$ – approximately equal to the right and left vectors of the largest singular value, and approximate $\sigma_{\max} \approx u^T W v$. The forward pass becomes:

$$y = \frac{Wx}{\sigma_{\max}(W)} \approx \frac{Wx}{u^T W v}$$

For this to work properly, the vectors $u$ and $v$ should be close to the vectors corresponding to the largest singular value. This is achieved via the power method [46, 47] by taking:

$$u \leftarrow W^T u / \|W^T u\| \qquad v \leftarrow Wv / \|Wv\|$$

By repeating this procedure for all layers, we ensure that the spectral norms of all layers are no larger than one. If that is the case, eq. (5) suggests that the critic should be stable in the forward pass. This would then bound the gradients being propagated into the actor as per eq. (3). Figure 3 has shown that exploding gradients are associated with poor performance, and we thus hypothesize that applying spectral normalization should stabilize learning and enable deeper networks.

## 5   Experiments

### 5.1   Smoothness Enables Larger Networks

We first investigate whether smoothing with spectral normalization allows us to use deeper modern networks. To do this, we simply compare using the modern network defined in Section 3.2 without and with spectral normalization. Specifically, for both the actor and the critic, we apply spectral normalization to each linear layer except the first and last. Otherwise, the setup follows Section 3.1. As before, when learning crashes, we simply use the performance recorded before crashes for future time steps. Learning curves for individual environments are given in Figure 4, again over 10 seeds. We see that after smoothing with spectral normalization, performance is relatively stable across tasks, even when using a deep network with normalization and skip connections. On the other hand, without smoothing, learning is slow and sometimes fails. Note that the improvements differ by tasks, but performance essentially improves monotonically when making the network deeper — just as in supervised learning. The only exception is walker walk, where the smoothed strategy is narrowly beat, however, this might be a statistical outlier. In Appendix A, we also show that smoothing with spectral normalization improves performance when using the 4 hidden layer MLPs. Thus, we conclude that enforcing smoothness allows us to utilize deeper networks.

### 5.2   Comparing to Other Agents

After demonstrating that smoothing with spectral normalization allows actor-critic agents to use larger modern networks which improve performance, we now see if these gains allow us to improve upon other agents. We compare against two state-of-the-art methods for continuous control: Dreamer [26] and the SAC-based agent of Kostrikov et al. [35] (DRQ), from which we start, using its default MLP architecture. To ensure fair and identical evaluation across different algorithms, we run DRQ and Dreamer with the author-provided implementations across 10 seeds. We show scores at step 500,000, averaged across 10 seeds in Table 2. For most tasks, the agent using deep modern networks with smoothing outperforms the other agents, although for a handful of tasks it is narrowly beaten. Note that the improvements that are obtained from scaling the network differ significantly between tasks, likely as some tasks are more complex and thus more amenable to high-capacity models. It is also interesting to note that performance on some sparse tasks improves, where artificial curiosity is often employed [54, 61], suggesting that such tasks might not always require specialized solutions. We note that these gains are comparable to those of algorithmic innovations, e.g. the improvement for changing architecture is larger than the difference between Dreamer and DRQ showed here, see Appendix D for details. Appendix D also compare against learning curves reported in [26, 35] which does not necessarily use more than 5 seeds. Here, deeper modern networks with smoothing again outperform the two alternatives. We conclude that by enforcing smoothness and simply scaling the network, it is possible to improve upon state-of-the-art methods without any algorithmic innovations.

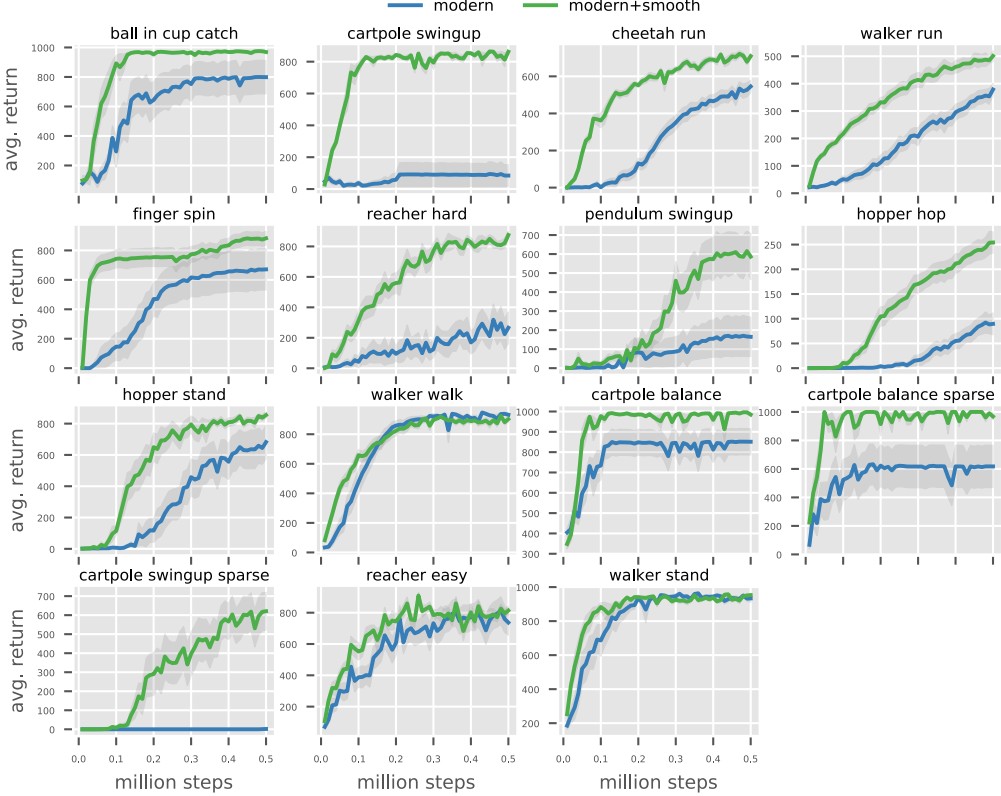

Figure 4: Learning curves when using deep modern networks, with and without smoothing by spectral normalization. Smoothing enables the deep network to learn effectively across tasks, whereas naively using such a deep network often leads to slow learning and sometimes no learning at all.

## 5.3 Experiments with DDPG

Our motivation for using spectral normalization is not specific to the SAC-based agent of Kostrikov et al. [35] and extends to actor-critic methods in general. To ensure that our methods generalize, we now consider the DDPG algorithm [43]. We evaluate how larger networks and spectral normalization affect the performance of the DDPG-based agent of Yarats et al. [71], dubbed DRQv2, and adopt hyperparameters therein. We use the same modern and base networks as for the SAC-based agent. For computational reasons we only focus on a smaller set of tasks, these include tasks with both dense and sparse rewards, and both locomotion and classical control tasks. Results over 10 seeds and after 500,000 frames are reported in Table 1. We see a similar result as earlier – using a larger network is detrimental to performance, but if spectral normalization is applied it can often be beneficial. These results suggest that our intuition for using spectral normalization benefits beyond the SAC setup. Results after 1 million frames are similar and are given in Appendix A. We conclude that spectral normalization can enable larger networks in DDPG too and that this agent also benefits from high-capacity networks.

## 5.4 Performance Cost

Scaling networks can improve performance, but larger networks can also increase memory footprint and compute time. Thus, we here measure the memory consumption and compute time for SAC-based agent of Kostrikov et al. [35] across four architectures; MLPs and modern networks (specified in Section 3.2) with two or four hidden layers. Recall that we only modify the head networks, which share features processed by a common convolutional network that is not modified. We consider training with batch size 512 and interacting with the environment, the latter set up models deployment after training. We use Tesla V100 GPUs and measure time with CUDA events and measure memory with PyTorch native tools. For a single run, we average time across 500 iterations with warm-start.

Table 1: Results for the DDPG agent of Yarats et al. [71] using different networks. Using a larger modern network results in poor performance compared to a traditional MLP, but once SN is used larger networks are beneficial. Metrics are given after 0.5 million frames and averaged over 10 seeds.

| task | MLP | modern | modern + SN |
|---|---|---|---|
| cup catch | $966.67 \pm 2.32$ | $107.46 \pm 21.43$ | $\mathbf{975.32} \pm 1.3$ |
| walker walk | $477.40 \pm 117.96$ | $6.10 \pm 9.08$ | $\mathbf{739.94} \pm 20.88$ |
| cartpole sparse | $\mathbf{1000.00} \pm 0.00$ | $11.27 \pm 0.16$ | $\mathbf{1000.00} \pm 0.00$ |
| hopper stand | $403.82 \pm 123.85$ | $2.95 \pm 0.99$ | $\mathbf{796.60} \pm 85.04$ |
| reacher easy | $756.15 \pm 70.48$ | $77.22 \pm 19.95$ | $\mathbf{801.83} \pm 66.31$ |

Table 2: Comparison of algorithms across tasks from the DeepMind control suite. We compare a modern architecture and spectral normalization (SN) [47] added to the SAC-based agent DRQ [35] against two state-of-the-art agents: DRQ with its default MLP architecture, and Dreamer [26]. By using spectral normalization the modern network outperforms the other methods. This demonstrates that simply stabilizing larger network can achieve gains comparable to those of algorithmic innovations.

| | ball in cup catch | swingup | cheetah run | walker run |
|---|---|---|---|---|
| MLP + DRQ | $884.6 \pm 262.0$ | $805.7 \pm 64.9$ | $502.8 \pm 315.5$ | $419.2 \pm 170.5$ |
| Modern+SN+DRQ | $\mathbf{968.9} \pm 13.3$ | $\mathbf{862.3} \pm 15.2$ | $\mathbf{708.9} \pm 46.2$ | $\mathbf{501.0} \pm 57.0$ |
| Dreamer | $767.1 \pm 63.3$ | $592.4 \pm 31.3$ | $630.7 \pm 23.3$ | $466.2 \pm 21.1$ |
| | finger spin | reacher hard | pendulum swingup | hopper hop |
| MLP + DRQ | $851.7 \pm 156.8$ | $674.7 \pm 332.0$ | $297.8 \pm 352.6$ | $196.1 \pm 72.0$ |
| Modern+SN+DRQ | $\mathbf{882.5} \pm 149.0$ | $\mathbf{875.1} \pm 75.5$ | $\mathbf{586.4} \pm 381.3$ | $\mathbf{254.2} \pm 67.7$ |
| Dreamer | $466.8 \pm 42.1$ | $65.7 \pm 16.5$ | $494.3 \pm 98.9$ | $136.0 \pm 28.5$ |
| | hopper stand | walker walk | balance | balance sparse |
| MLP + DRQ | $743.1 \pm 280.5$ | $754.3 \pm 356.7$ | $981.5 \pm 39.6$ | $910.5 \pm 268.6$ |
| Modern+SN+DRQ | $\mathbf{854.7} \pm 63.5$ | $\mathbf{902.2} \pm 67.4$ | $\mathbf{984.8} \pm 29.4$ | $\mathbf{968.1} \pm 72.3$ |
| Dreamer | $663.7 \pm 70.3$ | $863. \pm 27.2$ | $938.2 \pm 13.0$ | $935.2 \pm 22.5$ |
| | swingup sparse | reacher easy | walker stand | |
| MLP + DRQ | $148.7 \pm 299.3$ | $753.9 \pm 205.0$ | $696.9 \pm 329.4$ | |
| Modern+SN+DRQ | $\mathbf{620.8} \pm 311.0$ | $814.0 \pm 85.8$ | $953.0 \pm 19.8$ | |
| Dreamer | $267.8 \pm 25.4$ | $\mathbf{830.4} \pm 31.4$ | $\mathbf{955.3} \pm 8.7$ | |

The results averaged across 5 runs are given in Table 3. The memory consumption during training changes relatively little, increasing roughly $17\%$. The reason for this is of course that memory footprint comes both from the convolutional layers, which we do not modify, and the head networks, which we increase in size. As the convolutional activations are spread across spatial dimensions, these dominate the memory consumption, and increasing the scale of the head networks becomes relatively cheap. The memory consumption during acting, which is dominated by storing the model weights, changes dramatically in relative terms but is less than 200 MB in absolute terms. For compute time, we see similar trends. Increasing the scale of the head network is relatively cheap as processing the images dominates computation time. During acting, however, the increase in time is smaller, likely as the time is dominated by Python computation overhead on the CPU rather than GPU computation. This demonstrates that simply scaling the head networks can be relatively cheap for RL from pixels while significantly improving performance.

# 6 Related Work

Early work on deep RL primarily used simple feedforward networks, possibly processing images with a convolutional network [48, 49]. This is still a common strategy [35, 37, 39], and the setting we have focused on. For environments with sequential information, it can be fruitful to incorporate memory into the network architecture via RNNs [32] or Transformers [51, 52]. There is also work

Table 3: We compare memory and compute requirements on Tesla V100 GPUs for the SAC-based agent of Kostrikov et al. [35], using four different architectures for the heads. For training, increasing the capacity of the head incurs a relatively small cost in both memory and compute since convolutional processing is the bottleneck. Thus, the improvements of Table 2 are relatively cheap. During acting, memory cost increases a lot in relative terms, but less in absolute terms. Compute cost during acting changes little, likely as it is dominated by CPU rather than GPU computations.

|  | mlp2 | mlp4 | modern 2 | modern4 |
|---|---|---|---|---|
| train mem (GB) | $4.05 \pm 0.0$ | $4.19 \pm 0.0$ | $4.27 \pm 0.0$ | $4.73 \pm 0.0$ |
| act mem (MB) | $50.16 \pm 0.0$ | $93.26 \pm 0.0$ | $112.66 \pm 0.0$ | $247.04 \pm 0.0$ |
| train time (ms) | $98.14 \pm 0.15$ | $104.1 \pm 0.14$ | $105.76 \pm 0.23$ | $125.21 \pm 0.1$ |
| act time (ms) | $1.19 \pm 0.09$ | $1.26 \pm 0.02$ | $1.4 \pm 0.1$ | $1.35 \pm 0.02$ |

on how to utilize, but not necessarily scale, neural networks in RL, e.g. dueling heads [68] or double Q-learning [66]. The proposition that too-large networks perform poorly has been verified for policy gradient methods in the systematic study of Andrychowicz et al. [2]. There is also ample systematic work on generalization in RL [11, 18, 74] and many other relevant large-scale studies include [15, 30, 76]. Spectral normalization has been used to regularize model-based reinforcement learning by Yu et al. [72], however, the effects of SN seem to not be studied through ablations or controlled experiments here. Concurrently with our work, Gogianu et al. [21] has investigated the use of spectral normalization for DQN. In contrast with Gogianu et al. [21], we show how SN can be used to enable larger networks, and use it in RL algorithms where the actor takes gradients through the critic. This setting is fundamentally different from the discrete DQN setting, and one much closer to the GAN setup where SN has been developed. Sinha et al. [60] consider using dense connections to enable larger networks in RL. In contrast, we consider even larger models and also provide a solution that is agnostic to the underlying network architecture.

There are many proposed ideas for enforcing smoothness in neural networks, especially for GANs [5, 9, 22, 24, 57]. The idea of directly constraining the singular value has been investigated by multiple authors [19, 23, 47]. For GANs, one updates the weight of a generator network by taking the gradient *through* the discriminator network, similar to how the actor is updated in SAC by taking gradients *through* the output of the critic network. Due to this similarity, perhaps more techniques from the GAN literature apply to actor-critic methods in RL.

## 7 Discussion

**The Importance of Architecture.** Within RL, it is common to abstract the neural network as function approximators and instead focus on algorithmic questions such as loss functions. As we have shown, scaling up the network can have a dramatic effect on performance, which can be larger than the effects of algorithmic innovations. Our results highlight how low-level network architecture decisions should not be overlooked. There is likely further progress to be made by optimizing the architectures. Within supervised learning, there is ample systematic work on the effects of architectures [50, 64], and similar studies could be fruitful in RL. Whereas larger architecture requires more compute resources, this can potentially be offset by asynchronous training or other methods designed to accelerate RL [8, 16, 17, 45, 55].

**Evaluation in RL.** Supervised learning has already reached the state where performance seems to monotonically improve with network capacity [64]. As a consequence, it is common the compare methods with similar compute resources. We have demonstrated how architectural modifications can enable much larger networks to be used successfully in RL. If such benefits can be extended across agents, fair comparisons in RL might require listing the amount of compute resources used for novel methods. This is especially important for compute-intensive unsupervised methods [1, 39, 44, 59, 73, 75] or model-based learning [4, 12, 20, 28, 42].

**Limitations.** For compute reasons we have only studied the effects of networks on SAC and DDPG-based agents — these are state-of-the-art algorithms for continuous control underlying many popular algorithms – nonetheless, there are plenty of other algorithms. While our study gives some hints, an outstanding question is why large architectures are not popular in RL in general, beyond the

algorithms we consider. The idea of obtaining gradients through subnetworks is common and algorithms such as Dreamer [26] might also benefit from smoothing. Spectral normalization [47] is a relatively straightforward smoothing strategy and many alternatives which might perform better are known. We emphasize that the goal of our paper is to demonstrate that enabling large networks is an important and feasible problem, to not provide a solution for every conceivable RL algorithm. Finally, there are also further environments to try out. We have focused on continuous control from pixels as it is a setting relevant for real-world applications, but other common benchmarks such as Atari games [7] and board games [3] are also important.

**Conclusion.** We have investigated the effects of using modern networks with normalization and residual connections on SAC [25, 35]- and DDPG [43, 71]-based agents. Naively implementing such changes does not necessarily improve performance, and can lead to unstable training. To resolve this issue, we have proposed to enforce smoothing via spectral normalization [47]. We show that this fix enables stable training of modern networks, which can outperform state-of-the-art methods on a large set of continuous control tasks. This demonstrates that changing network architecture can be competitive with algorithmic innovations in RL.

## Acknowledgement and Funding Transparency

This research is supported in part by the grants from the National Science Foundation (III-1618134, III-1526012, IIS1149882, IIS-1724282, and TRIPODS- 1740822), the Office of Naval Research DOD (N00014- 17-1-2175), Bill and Melinda Gates Foundation. We are thankful for generous support by Zillow and SAP America Inc. This material is based upon work supported by the National Science Foundation under Grant Number CCF-1522054. We are also grateful for support through grant AFOSR-MURI (FA9550-18-1-0136). We also acknowledge support from the TTS foundation. Any opinions, findings, conclusions, or recommendations expressed here are those of the authors and do not necessarily reflect the views of the sponsors. We thank Rich Bernstein, Joshua Fan, Ziwei Liu, and the anonymous reviewers for their help with the manuscript.

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
