# A Appendix

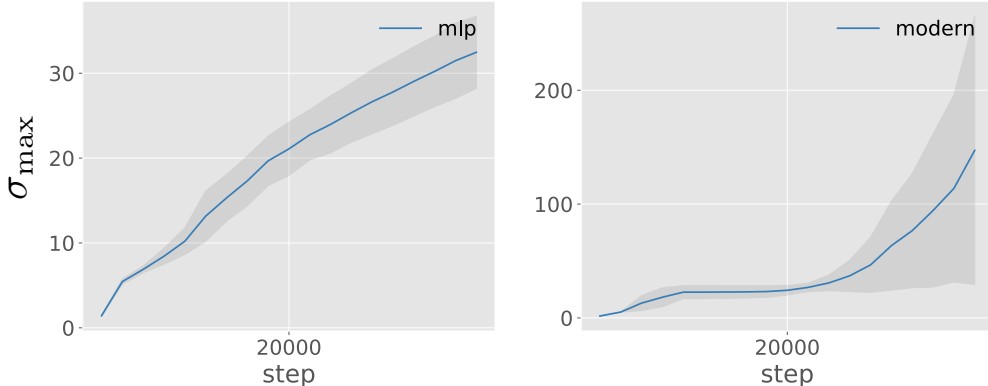

Figure 5: The largest singular value of the weight matrix at layer 2 when using the MLP or modern architecture (without smoothing). Results are shown for the cartpole swingup environment and averaged across 5 runs. For the modern network with skip connections and normalization, the largest singular value grows dramatically.

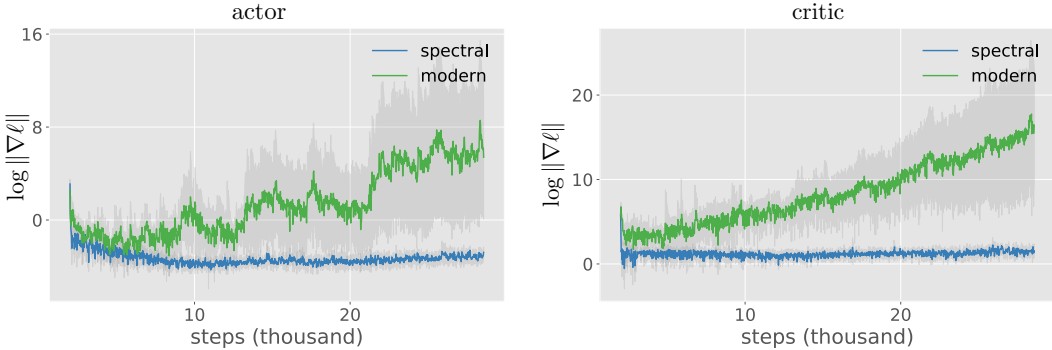

Figure 6: We here compare the gradients when using a modern network with and without smoothing with spectral normalization. When using spectral normalization, the gradients are stable during training even for the larger modern networks. Compare with Figure 3.

# B Hyperparameters list

Hyperparameters are listed in Table 4.

# C Network Details

The network details follow Kostrikov et al. [35] except for the heads. The image is resized into an 84-by-84 tensor with frame-stacking from the last 3 steps. This tensor is then processed by the convolutional encoder which consists of four convolutional layers with 3-by-3 kernels and ReLu between them. The encoder uses 32 filters and stride 1 except for the first layer, which uses stride 2. This spatial map is then fed into a linear layer which outputs a 50-dimensional vector, which is processed by a layer norm [6] unit. The MLPs for the critic and actor are feedforward networks that alternate between linear transformations and Relu, using a width of 1024 and two hidden layers.

For the modern architecture we use the transformer feedforward architecture [67] with layer norm applied before the input; i.e., we have $x + w_2\text{Relu}(w_1\text{norm}(x))$. Dropout is not used. The width of the residual branch is 1024 whereas the linear transformations upscale and downscale the width to 2048. The network will need to increase the initial feature dimension to 1024 and in the final layer

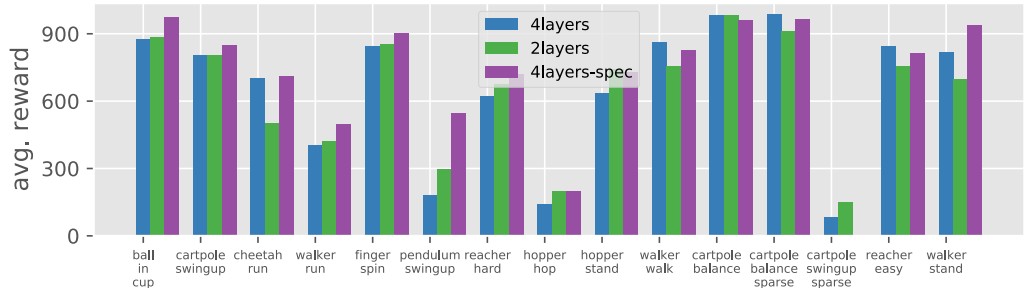

Figure 7: The performance when making the networks deeper. Performance is averaged across ten seeds and plotted for 15 tasks from the DeepMind control suite [65]. When smoothing with spectral normalization [47], the performance improves for the 4-hidden-layer network for most tasks.

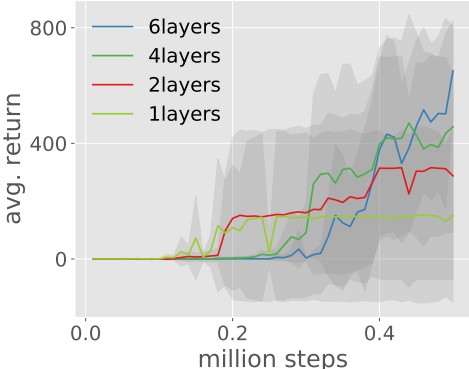

Figure 8: Small ablation experiment on cartpole swingup sparse where we vary the number of *modern* network blocks in both the critic and the actor while smoothing with spectral normalization. Performance is average across 5 seeds. Performance seems to increase, or at least not decrease, when using deeper networks.

decrease it to 1 or twice the number of actions. To this end, one linear layer is used at either end of the network. For the purpose of calculating depth, we count these two linear layers as one transformer layer (since they both use two linear layers). When using spectral normalization, the normalization is added to all *intermediate* layers — i.e. not the first and last layer. The reason for this is that we do not want the network to be 1-Lipschitz throughout, as we might need them to represent very large q-values.

## D  Improvements

As per Table 2, the average improvement when changing architecture from the 2 hidden layer MLP to the modern network is slightly more than 141. We can compare this to improvements gained from algorithmic innovations between the Dreamer [26] and DRQ agent [35] which Table 2 puts at less than 40. We can also compare this to the improvements gained for CURL [39], which represented a new SOTA at the time of publication. Specifically, we consult Table 1 of the CURL paper [39]. There, performance is also measured after 500,000 steps on DeepMind control suite [65], however a smaller set of environments is used. CURL reaches an average score of 846 whereas the previously published Dreamer agent [26] scores 783. This results in an improvement of 63. In follow-up work of [37], the average scores increase to 898, which is an even smaller improvement.

In Table 5 we compare using deeper networks smoothed with spectral normalization against Dreamer [26] and DRQ [35], using the curves from [26, 35]. Dreamer is there run with only 5 seeds [26]. For the sake of using 10 seeds uniformly and controlling for changes in the DRQ setup, we run the baselines ourselves with code released by the authors [26, 35] in Table 2. Since we modify the

Table 4: Hyper-parameters for the experiments, following [35]. We use action repeat of 2 for the dreamer benchmark, following [26, 35].

| Parameter | Value |
|---|---|
| $\gamma$ | 0.99 |
| $T_0$ | 0.1 |
| $\tau$ | 0.01 |
| $\alpha_{adam}$ | 1e-3 |
| $\epsilon_{adam}$ | 1e-8 |
| $\beta_1 adam$ | 0.9 |
| $\beta_2 adam$ | 0.999 |
| batch size | 512 |
| target update freq | 2 |
| seed steps | 5000 |
| $\log \sigma$ bounds | [-10, 2] |
| actor update frequency | 2 |
| seed steps | 1000 |
| action repeat | 2 |

Table 5: Comparison of algorithms across tasks from the DeepMind control suite. We compare a modern architecture and spectral normalization (SN) [47] added to the SAC-based agent DRQ [35] against two state-of-the-art agents: DRQ with its default MLP architecture, and Dreamer [26]. Here we use numbers for [26, 35] for DRQ and Dreamer obtained through communications with the authors. Note that these do not use 10 seeds consistently, and some tasks thus appear to have an artificially low standard deviation. Using modern networks and smoothing still comes out ahead, beating dreamer at 11 out of 15 games and beating MLP at 13 out of 15 games.

| | ball in cup catch | swingup | cheetah run | walker run |
|---|---|---|---|---|
| MLP + DRQ | $945.9 \pm 20.7$ | $825.5 \pm 18.2$ | $701.7 \pm 22.1$ | $465.1 \pm 27.2$ |
| Modern+SN+DRQ | $\mathbf{968.9} \pm 13.3$ | $\mathbf{862.3} \pm 15.2$ | $\mathbf{708.9} \pm 46.2$ | $\mathbf{501.0} \pm 57.0$ |
| Dreamer | $961.7 \pm 2.2$ | $678.4 \pm 108.2$ | $630.0 \pm 16.1$ | $463.5 \pm 37.0$ |

| | finger spin | reacher hard | pendulum swingup | hopper hop |
|---|---|---|---|---|
| MLP + DRQ | $\mathbf{905.7} \pm 43.2$ | $824.0 \pm 35.7$ | $584.3 \pm 114.5$ | $208.3 \pm 11.5$ |
| Modern+SN+DRQ | $882.5 \pm 149.0$ | $\mathbf{875.1} \pm 75.5$ | $586.4 \pm 381.3$ | $\mathbf{254.2} \pm 67.7$ |
| Dreamer | $338.6 \pm 24.3$ | $95.2 \pm 26.1$ | $\mathbf{789.2} \pm 25.7$ | $120.0 \pm 18.5$ |

| | hopper stand | walker walk | balance | balance sparse |
|---|---|---|---|---|
| MLP + DRQ | $755.9 \pm 39.7$ | $897.5 \pm 24.3$ | $\mathbf{993.5} \pm 2.7$ | $903.2 \pm 63.4$ |
| Modern+SN+DRQ | $\mathbf{854.7} \pm 63.5$ | $\mathbf{902.2} \pm 67.4$ | $984.8 \pm 29.4$ | $968.1 \pm 72.3$ |
| Dreamer | $609.7 \pm 76.2$ | $885.8 \pm 12.6$ | $981.8 \pm 3.1$ | $\mathbf{997.7} \pm \mathbf{0.9}$ |

| | swingup sparse | reacher easy | walker stand | |
|---|---|---|---|---|
| MLP + DRQ | $520.4 \pm 114.1$ | $804.8 \pm 31.0$ | $944.4 \pm 17.8$ | |
| Modern+SN+DRQ | $620.8 \pm 311.0$ | $\mathbf{814.0} \pm 85.8$ | $953.0 \pm 19.8$ | |
| Dreamer | $\mathbf{799.7} \pm 7.4$ | $429.5 \pm 82.3$ | $\mathbf{960.0} \pm 7.0$ | |

architecture and source code of DRQ, running experiments with 2 hidden layer MLPs ourselves allows us to make sure that the setup is identical – even details such as CUDA versions, minor code changes, GPUs, and seeds can have a large effect in RL [30]. Our setup of modern networks and smoothing beats Dreamer for 11 out of 15 games and beats the 2-hidden-layer MLP of DRQ for 13 out of 15 games in Table 5. At any rate, we do not make any specific claims about what algorithms might be the best — only that one can improve control from pixels by using deeper networks with smoothing.

Table 6: Results for the DDPG-based agent of Yarats et al. [71] after 1 million frames. Using a larger modern network results in poor performance compared to a traditional MLP, but once SN is used larger networks are beneficial. Metrics are given over 10 seeds.

| task | MLP | modern | modern + SN |
|---|---|---|---|
| cup catch | $972.76 \pm 2.03$ | $105.58 \pm 27.65$ | $978.89 \pm 1.73$ |
| walker walk | $581.57 \pm 143.06$ | $57.29 \pm 25.95$ | $958.07 \pm 2.32$ |
| cartpole sparse | $999.54 \pm 0.44$ | $11.31 \pm 0.13$ | $1000.00 \pm 0.00$ |
| hopper stand | $567.84 \pm 137.13$ | $4.39 \pm 1.28$ | $832.01 \pm 86.13$ |
| reacher easy | $929.52 \pm 15.71$ | $86.60 \pm 15.56$ | $957.29 \pm 11.7$ |

## E  Compute Resources

Experiments were conducted with Nvidia Tesla V100 GPUs using CUDA 11.0 and CUDNN 8.0.0.5, running on nodes with Intel Xeon CPUs. Experiments contained in the paper represent approximately a year or two of GPU time.