# OpenReview forum: "Towards Deeper Deep Reinforcement Learning with Spectral Normalization"
_NeurIPS.cc/2021/Conference — NeurIPS 2021 Poster_

### Official Review · Reviewer_mJA7 · 2021-07-09

**Rating:** 7
**Confidence:** 4

**Summary:**

The paper hypothesizes that the poor performance of modern NN architectures with the SAC algorithm stems from large gradient updates caused by propagating gradients for actor through critic. The authors mitigate the issue by borrowing a spectral normalization technique from GANs, which was designed for a similar problem. Experimentally, they verify that the approach improves the performance and allows usage of modern architectures, with gains comparable to algorithmic advances of RL algorithms themselves.

**Limitations And Societal Impact:**

- addressed limitations
- no apparent negative societal impact

**Main Review:**

The paper is well and systematically written. It gradually presents the main problem - that modern architectures do not work well with the SAC algorithms, tests it accordingly, and rule out the overfitting hypothesis. It then presents the main culprit - large gradient updates caused by taking gradients through critic, and theoretically shows that spectral normalization can help. Experimentally, they verify that it is indeed the case.

Although the actual improvement is trivial - just use a spectral normalization - the potential impact of the results is high. The experiments show that the gains from the technique is comparable to algorithmic improvements. Also, the paper does not simply report a result, but it clearly explains the underlying cause of the problem.

A limitation is that the issue is only present in SAC-similar algorithms that propagate the gradient through critic. Nevertheless, it also needs to be said that this class of algorithms is important and well used. The authors are aware of the limitation and state it clearly.

Notes:
- l. 169 - unclear whether $f$ in $||f||$ is a substitute symbol for any function or the already defined $f$ symbol.
- Table 1 - it would strengthen the paper and be interesting to see how the spectral normalization help in the DRQ / DDPG / Dreamer algorithms
- similarly, it would be interesting to see whether the effect helps in other classes of algorithms (or not), where the gradient is not taken through critic

**Time Spent Reviewing:**

5

---

> ### Author Response · Authors · 2021-08-10
> **Rebuttal**
>
> We are thankful for the comments and the positive response from the reviewer!
>
> 1. *l. 169 - unclear whether f in ||f|| is a substitute symbol for any function or the already defined f symbol.*
>
> f is the previously defined symbol for the activation function, we will clarify.
>
> 2. *Table 1 - it would strengthen the paper and be interesting to see how the spectral normalization help in the DRQ / DDPG / Dreamer algorithms*
>
> This is a good direction for future research, we hope to try more RL algorithms!

---

### Official Review · Reviewer_FNPT · 2021-07-09

**Rating:** 6
**Confidence:** 4

**Summary:**

The paper starts from the following observation: Deeper and wider architectures usually provide better performance in supervised learning. This is often not the case in Reinforcement learning.

The paper then proceeds to formulate two hypothesis for this observation: deeper architectures could overfit, or deeper architectures could be harder to train, due to ill-conditioning of gradients. After refuting the first hypothesis, the paper provides a theoretical argument for the second, and provide an attempt at fixing it using spectral normalization. The paper provides improved experimental results with deeper architectures using spectral normalization.

**Limitations And Societal Impact:**

The authors adequately addressed the limitations and potential negative societal impact.

**Main Review:**

The paper at hand provides an interesting and simple solution which seems to enable the use of deeper architectures in Reinforcement Learning. The paper reads well, and empirical results are compelling and well presented.

However, I have a few concerns with the current revision:
- My first concern is about the novelty and originality of the provided approach. Applying Spectral Normalization to help optimization in an RL context has already been analyzed in (Gogianu et al. 2021), with similar conclusions, though on a different base algorithm. This reference is unfortunately missing in the current draft. Applying the same technique to a different algorithm and different environments makes the contribution of this paper somewhat incremental.

Edit: The Area Chair has brought to my attention that (Gogianu et al. 2021) was only available online on arxiv within the two months interval before the ICML submission, and that it should thus be considered concurrent work. I've edited my score accordingly, but still think that this concurrent submission should be discussed in the final revision.

- My second concern is about the soundness of the theoretical analysis. In the introduction, the authors mention that scaling up the critic typically lead to exploding gradients (which seems empirically true), but mention that they are motivating it mathematically. However, the theoretical analysis provided in the paper gives an upper bound for the gradient of the critic. This upper bound does not give proper backing for the exploding gradient hypothesis, a lower bound would be much better suited.
- My third, and probably lesser concern is about the first hypothesis mentioned by the paper, that bigger networks could lead to more overfitting in simple reinforcement learning environments. While this could well be a valid hypothesis, the phrasing of the paper seems to imply that this hypothesis is backed up by the relative scale of reinforcement learning 'datasets' compared to supervised datasets. I don't think this holds in general, and I think that in many instances, the 'datasets', referring to the total number of unique observations that the agent has seen during training is of the same order of magnitude, or higher than the number of datapoints considered in supervised settings (e.g. A3C trains on roughly 160 millions frames total).

As the paper is currently mostly experimental, I think a direction that would significantly improve the paper would be to test the hypothesis that the architectural change performed improves performance across RL algorithms and environments. While I understand that this is both computationally costly, and far from trivial to implement, this would provide a much stronger and general result.

For the reasons mentionned above, I think the paper is not yet ready for publication at NeurIPS.

--------
References:

_Spectral Normalisation for Deep Reinforcement Learning: An Optimisation Perspective_ Florin Gogianu, Tudor Berariu, Mihaela Rosca, Claudia Clopath, Lucian Busoniu, Razvan Pascanu, ICML 2021

# Post rebuttal and additional experiments edit:
The authors have provided further results on DDPG, which was one of the things that I asked for. I think those results already make the paper a lot stronger. For this reason, I am raising my score from 4 to 5. The reason why I am not increasing it further is because I am still concerned by the theoretical part, and the claims associated to it. While I agree that the bounds can be made tight, I don't think it strengthen or back up the claims made in the introduction. I would suggest removing this theoretical part, and only backing the claims up experimentally. Removing this part would also probably allow the authors to put the results on DDPG in the main text.

# Post rebuttal second edit:
The authors have agreed to remove the theoretical part that I considered potentially misleading. I no longer have any major concerns about the paper, and I am raising my score from 5 to 6.

**Time Spent Reviewing:**

3

---

> ### Comment · Area_Chair_iY2m · 2021-08-04
> **On Concurrent Work**
>
> Dear Reviewer FNPT,
>
> Thank you very much for your efforts in contributing to the NeurIPS 2021 reviewing process by reviewing this paper.
>
> The reviewer had concerns regarding the novelty and originality of the provided approach. The review suggested that the application of Spectral Normalization to help optimization was already analyzed in (Gogianu et al. 2021) with similar conclusion but with a different base algorithm which is correct. However, (Gogianu et al. 2021) was put on arxiv just a few weeks before the NeurIPS deadline. At NeurIPS, the papers appearing less than two months before the submission deadline are generally considered concurrent to NeurIPS submissions. Due to the relevancy of (Gogianu et al. 2021) to this submission, I would advise the reviewer to ask the authors to acknowledge and discuss the other paper in the camera-ready version if the submission is accepted.
>
> I would advise the reviewer to update their review not to judge the paper's novelty based on (Gogianu et al. 2021).
>
> Thanks!

---

> ### Author Response · Authors · 2021-08-10
> **Rebuttal**
>
> We are thankful for the comments and directions for future research. In light of the AC comments, we hope that the reviewer will update the review.
>
> 1. *My first concern is about the novelty and originality of the provided approach. Applying Spectral Normalization to help optimization in an RL context has already been analyzed in (Gogianu et al. 2021), with similar conclusions, though on a different base algorithm.*
>
> The paper of Gogianu et al. was put on arXiv on the 11th of May, (and of course published at ICML even later) whereas the deadline for Neurips was May 28th. The difference is thus 2.5 weeks, and we have conducted our research independently of this publication. NeurIPS has a policy of considering work made public within 2 months concurrent -- please see https://neurips.cc/Conferences/2021/PaperInformation/NeurIPS-FAQ. We will cite and discuss this paper. In contrast with Gogianu, we show how SN can be used to enable larger networks, and use it in RL algorithms where the actor takes gradients through the critic. This is a fundamentally different setting, and one much closer to the GAN setup where SN has been developed. We believe that both papers have distinct contributions, and hope that the reviewer will follow the recommendation of the AC and not judge our work's novelty based upon a paper posted on arXiv 2.5 weeks before the submission deadline.
>
> 2. *This upper bound does not give proper backing for the exploding gradient hypothesis, a lower bound would be much better suited.*
>
> This is a subtle but important issue, thanks for bringing this up! In general, it is impossible to provide a non-zero lower bound because the gradient can be zero if e.g. all activations are in the ‘dead’ region of a Relu function through which gradients do not propagate. However, all inequalities we use to derive the bound can be made tight for some combinations of weights and inputs, so in principle, our bound can be tight for very specific inputs/weights. We will comment on this. Note that our bound is in line with our experimental results in Figures 3 and 6.
>
>
> 3. *in many instances, the 'datasets', referring to the total number of unique observations that the agent has seen during training is of the same order of magnitude, or higher than the number of datapoints considered in supervised settings (e.g. A3C trains on roughly 160 millions frames total).*
>
> In this paper we consider sample-efficient RL, where one can solve most dm-control tasks in less than a million frames -- we are thus far apart from the 160 million frames of A3C. Furthermore, as stated on line 32, another issue is that the diversity of frames obtained in RL likely are much more redundant than in e.g. Imagenetnet.
>
> 4. *I think a direction that would significantly improve the paper would be to test the hypothesis that the architectural change performed improves performance across RL algorithms and environments.*
>
> This is a good direction for future research, we hope to try one more RL algorithm!

---

> ### Author Response · Authors · 2021-08-29
> **Response : Post rebuttal and additional experiments edit:**
>
> Thanks for the updated review!
>
> The empirical results are likely the main attraction of the paper. The theoretical results (i.e. proposition 1) are relatively minor, while providing some intuition they might not sufficiently strengthen our experimental results due to being upper bounds. This perspective appears to be shared by reviewer vkHq. Thus, it appears the theoretical results mainly distract from the positive empirical results. Hence, following your recommendation, we will remove the theoretical results/claims from the submission and focus on empirical results. As stated, this would allow more space that can be dedicated to the recent experiments with DDPG. Thanks for the fruitful discussions and concrete tips for improving the manuscript!

---

### Official Review · Reviewer_vkHq · 2021-07-16

**Rating:** 6
**Confidence:** 4

**Summary:**

The paper argues that network architectures are under explored in reinforcement learning and that much can be gained from understanding the lack of success of larger models. The paper focuses on continuous control from pixels with the SAC algorithm and starts by demonstrating that a naive application of ‘modern’ architectures leads to a degradation in performance. The authors argue that this degradation in performance is at least partially due to instabilities caused by large gradients from the critic to the policy actions. They provide a mathematical motivation for why this is a bigger problem for deeper networks and propose to apply spectral normalization, which is a popular technique for controlling the gradients of the discriminators in GANs. Finally the paper reports empirical results supporting the claim that this method allows for the use of larger architectures and by virtue of that SOTA results that are on par or better than results obtained with algorithmic innovations.


**Limitations And Societal Impact:**

Yes.

**Main Review:**

The novelty of the work is limited. As the authors point out, spectral normalization has been applied to critic gradients before. To my knowledge this is however the first time that its effect on learning performance is investigated systematically in relation to larger networks. The notion that larger networks are not as prominent in reinforcement learning is not new either (neither do the authors claim this) but excluding overfitting as the explanation and investigating why this is the case for the set of critic-gradient using RL algorithms could be novel.

The largest problem I see with this paper is that at various parts it promises to address why large architectures are not popular for use with RL algorithms in general while focussing on an issue that only affects those algorithms that use the gradient of the critic with respect to the actions. Examples of such algorithms are DDPG/TD3 and SAC, which are popular in continuous control but arguably still not as commonly used as Q learning, PPO and TRPO, which don’t use critic gradients. The authors mention this limitation and write that their goal is not to “provide a solution for every conceivable RL algorithm”. They are right that this is of course too much to expect from a single paper but by addressing an issue that is caused by a property that only affects a small number of algorithms, they don’t investigate why small networks are not popular with RL in general like the abstract and introduction suggest, especially considering that critic gradient using algorithms are relatively new compared to standard policy gradient and Q-learning.

The experiments are done on a variety of tasks from a well-known benchmark suite and this adds to the quality of the paper. The results are also the averages of multiple runs in most case but for the results in Figure 1 and 2 this is not clear as there are no error bars.
The evaluation of the critic loss in 3.4 is somewhat hand-wavy because it cannot be considered in isolation of what the actor is doing. The critic loss could also be higher because a larger variety of states and returns has been observed due to increased exploration for example.

The paper is a bit messy at times and this makes some results a bit harder to interpret and some of the mathematics a bit confusing or simply incorrect, especially in the appendix.
The baseline DRQ/mlp results in both Table 1 and Table 4 (appendix) seem to be quite a bit worse than the numbers reported in Kostrikov et al. (2020) [34] Table 1 and it is not clear to me where those numbers come from. The appendix says that they come from the curves in the original papers [25,34] but I don’t understand how precise mean estimates and precision values (not clear from the captions whether these are standard deviations or standard errors) can be obtained from those. It seems to me like the current paper uses the same control suite tasks but not he exact same PlaNet evaluation setup but this is not 100% clear either. Having baseline results that are apparently much worse than those reported in the original papers undermines all the comparisons with papers and is very misleading.

Here are some examples of issues with the mathematics:
- It seems to me that the maximization in Equation 4 (and other equations derived from it) should be a minimization (or otherwise placed before the fraction) because as it stands, the norm of the total gradient is now upper-bounded by sqrt(N) times the value corresponding to the layer with the lowest norm bound. This seems to replace an upper-bound on a (square rooted) sum with a repetition of its lowest term, which must be wrong (the bound on the total gradient would become lower than the bounds on some of the individual layers if we let the operator norm of one of the layers grow much larger than the others).
- The `A` in the equation at the end of Section 4.2 should probably be a ‘Q’ because the gradient magnitude of the actor output itself doesn’t depend on the critic weights.
- The subscripts of the products in Equation 6 of the appendix don’t make sense to me. I assume they should have values like ‘j=i+1’ instead of ‘j+1’
- It’s a bit odd that both the actor and critic have N layers, making it less clear what the result would be when these numbers were different.
- Last sentence of p16 (appendix): “one obtains an estimate of the output of the j:th layer:” is followed by Equation 6 which clearly defines bounds on the norms of the layer outputs and not estimates of those layers.
- Second sentence p17 (appendix): No superscript on the weight and ‘Q’ should probably be ‘A’.

Another point that was not clear from the text but that is implicitly addressed in the appendix is that the arguments about the spectral normalization only make sense if the spectral norms of larger nets also grow during learning. Otherwise, initialization with small enough weights for deeper nets would be an easier solution than spectral normalization.

Edit: I raised my score in light of the clarifications by the authors and the additional DDPG results.


**Time Spent Reviewing:**

6

---

> ### Author Response · Authors · 2021-08-10
> **Rebuttal**
>
> We thank the reviewer for a thoughtful and detailed review. We are grateful for the suggested improvements to the theoretical parts which we will fix/clarify. Both critic and actor have N layers for ease of exposition, this is not a fundamental issue with the proof. We will also clarify the indices in (eq 6) and clarify how it relates to bounding the layer outputs. Further questions are answered individually below:
>
>
>
> 1. *The novelty of the work is limited. As the authors point out, spectral normalization has been applied to critic gradients before.*
>
> We respectfully disagree, the only earlier work we are aware of which applies spectral normalization (SN) in RL is (Mopo: Model-based offline policy optimization. T. Yu, et. al.). This paper only mentions spectral normalization *once* in the appendix. Furthermore, it does not study the effects of applying spectral normalization, i.e. there are no ablation experiments for SN and no motivation as to why SN is used. Thus, while spectral normalization has been used once before, we argue that it has not been studied. Is the reviewer aware of any further earlier work?
>
>
>
> 2. *The largest problem I see with this paper is that at various parts it promises to address why large architectures are not popular for use with RL algorithms in general.*
>
> This seems like a misunderstanding. We do not promise to address why large architectures are not popular in RL across all algorithms. This is a larger question outside the scope of our paper. We focus exclusively on RL algorithms where gradients are taken through the critic. To contextualize and motivate why our focus is interesting, we discuss the general state in RL. We will clarify this.
>
>
>
> 3. *The results are also the averages of multiple runs in most case but for the results in Figure 1 and 2 this is not clear as there are no error bars.*
>
> The results in Figures 1 and 2 are averaged over 10 runs. We will add error bars to these figures to clarify.
>
>
>
> 4. *It seems to me like the current paper uses the same control suite tasks but not the exact same PlaNet evaluation setup but this is not 100% clear either.*
>
> Please note that we are NOT using the planet evaluation setup. Firstly, the planet evaluation setup uses 6 tasks, we consider 15. Secondly, the planet setup uses action repeats that are tuned individually for each environment, whereas we consider an action repeat of 2 uniformly across tasks. The numbers we obtain are expected to be lower than those that can be obtained with tuned action repeats. Our setup follows the setup of Figure 4 in (2020, Kostrikov et. al.). We specify our setup on lines 103-104 but will clarify this further.
>
>
>
> 5. *The baseline DRQ/mlp results in both Table 1 and Table 4 (appendix) seem to be quite a bit worse than the numbers reported in Kostrikov et al. (2020)*
>
> The reviewer seems to be mistaken here. The numbers from Table 4 are taken straight from (Kostrikov et. al.), we have emailed the authors to obtain these. Thus, they cannot be worse. As explained above, we do not consider the planet setup so the reviewer should not compare the results in our paper to those which Kostrikov et. al obtains for the Planet benchmark. The major difference between tables 1 and 4 comes from the dreamer agent -- the numbers in table 4 are computed across only 5 seeds which gives artificially low standard deviations for a few tasks, see e.g. ball in cup catch. We will clarify that our setup differs from the Planet benchmark and merge tables 1 and 4 to avoid confusion.
>
>
>
> 6. *The maximization in Equation 4 (and other equations derived from it) should be a minimization*
>
> Yes, we will exchange the max for a min. Thanks for catching this!
>
>
>
> 7. *The A in the equation at the end of Section 4.2 should probably be a ‘Q’*
>
> It is supposed to be a Q(A), we will fix this!
>
>
> 8. *[SAC/DDPG type algorithms are] arguably still not as commonly used as Q learning, PPO and TRPO, which don’t use critic gradients*
>
> Due to spectral normalization’s connections with GAN, we focus on SAC-style algorithms that have been developed for continuous control. For the Deepmind control suite (arguably the most popular continuous control benchmark), the vast majority of SOTA algorithms developed recently (CURL, RAD, DRQ) relies on SAC. Thus the class we consider is important for practical applications. SAC of course has roots in q-learning, so implicitly we are investigating q-learning. “Raw” q-learning with discretized actions is not very common for continuous control. We will clarify our scope in the main text.
>
>
> 9. *Another point that was not clear from the text but that is implicitly addressed in the appendix is that the arguments about the spectral normalization only make sense if the spectral norms of larger nets also grow during learning. Otherwise, initialization with small enough weights for deeper nets would be an easier solution than spectral normalization.*
>
> We will clarify this in the main text.  As per Figure 5, the spectral norm does grow for larger networks.
>
> Edits: we added answers to points 8 and 9.

---

### Author Response · Authors · 2021-08-29
**Update**

We thank the reviewers for fruitful discussions! Following suggestions from reviewer FNPT, we have decided to remove the theoretical results (i.e. proposition 1), which are relatively small, from the submission and instead focus only on empirical results. The strongest point of our work is likely the experimental results, and the theoretical claims might act more as a distraction, in part due to the theoretical results being upper bounds rather than lower bounds. Removing the theoretical results would also free up space to add our recent results on DDPG to the main paper.

At this point, we believe that most issues brought up in the reviews are addressed. Reviewer vkHq has voiced some issues, but we believe that these are addressed. Importantly, we have addressed points in review vkHq by:

* 1. Adding experiments with DDPG to show that our methods are broadly applicable, and committed to clarifying our scope -- using larger networks for SAC-style algorithms on continuous control tasks.
* 2. Committing to removing the minor theoretical results, which should address issues brought up by vkHq regarding the theoretical exposition.
* 3. Clarifying how our setup differs from the planet benchmark, which likely is the cause of confusion for the baselines we use.
* 4. Committing to adding error bars to Figures 1 and 2 (the means are already calculated over 10 seeds).

We would like to ask reviewer vkHq if there are issues we can address further or if any additional clarifications are needed? Thanks!

---

> ### Comment · Reviewer_vkHq · 2021-08-30
> **Raised my score and have a question about gradient norm clipping**
>
> I thank the authors for all the clarifications and addressing many of my concerns.
>
> The misunderstanding about the exact benchmark used was unfortunate but I hope that the paper will also mention that the baseline numbers were obtained through personal communication to avoid future confusion. This was the main reason that I raised my score previously. I agree that the paper will probably be stronger without the analytic part as well.
>
> An important concern is currently that it seems like the amount of rewriting would potentially warrant another round of review but with the new DDPG results I decided to raise my score regardless.
>
> One of the reasons why I don't see the insight of unstable gradients in gradient-based AC methods as that novel (and that I regret not pointing out earlier) is that implementations of DDPG in popular frameworks on github like OpenAI's baselines and Deepmind's ACME come with a gradient norm clipping options by default. This is not a fatal issue by itself because I haven't found papers that focus on this and the observed connection with network size and general empirical results are still valid contributions. However, now that DDPG results have been added, I'd be very curious to know if experiments with gradient norm clipping were also done or could still be done. If SN is on par with gradient norm clipping, I don't consider that a real issue but if SN works (much) better than clipping or is at least easier to tune, that could increase the impact of the paper significantly.

---

> > ### Author Response · Authors · 2021-08-30
> > **Thanks for the feedback, we will start experiments on clipping!**
> >
> > We thank reviewer vkHq for thoughtful feedback and for updating the review!
> >
> > We will make sure to mention that we have obtained baseline numbers through private communications to avoid confusion. It pleases us that there is consensus that removing the minor theoretical claim (proposition 1) will make the paper stronger. This is an easy fix that opens up some space for the DDPG experiments. We haven’t experimented with gradient clipping but will start such experiments! However, with limited GPUs these are unlikely to finish within the discussion period. Judging from the growing gradient magnitude in Figure 3, setting clipping thresholds appropriate throughout training might be hard, whereas SN requires no hyperparameter tuning.
> >
> >
> > ## Edit: results on the last day of discussion
> >
> > Given limited time, we have investigated gradient norm clipping on the DDPG agent (it runs more than 2x faster than the sac agent due to the base implementation being faster) on the small set of tasks consider in our previous update. Again we use the larger "modern" networks. We clip the gradient norm of the actor, critic, and encoder independently by using the same threshold. In the few days between the request and the end of the discussion period, we've only been able to use clipping values {10,100} and using {10,8} seeds for these respectively. These initial results do not show gradient clipping performing well, although this might be due to the hyperparameters (i.e the clipping value) being suboptimal. Thus, we don't consider these results conclusive but wanted to share what we have given that the discussion period ends today.
> >
> > ## Table 1. Performance at 0.5 million frames.
> >
> > | task			|	clip-10 		|	clip-100
> > | --- | ----------- | ------- |
> > | cup catch		|	108.67  $\pm$ 25.91	|	148.41 $\pm$ 24.67
> > | walker walk		|	23.12 $\pm$ 1.64 |		25.94 $\pm$ 1.98
> > | cartpole sparse	|	11.35 $\pm$ 0.15	|	65.36 $\pm$ 50.62
> > | hopper stand		|	2.54 $\pm$ 0.59 |	3.53 $\pm$ 1.16
> > | reacher easy		|	133.18 $\pm$ 70.12	|	108.06 $\pm$ 26.97
> >
> >
> > ## Table 2. Performance at 1.0 million frames.
> >
> > | task			|	clip-10 		|	clip-100
> > | --- | ----------- | ------- |
> > | cup catch		|	128.53 $\pm$ 28.13	|	98.06 $\pm$ 17.46
> > | walker walk		|	28.08 $\pm$ 1.45 |		28.39 $\pm$ 2.81
> > | cartpole sparse	|	11.26 $\pm$ 0.12	|	130.77 $\pm$ 111.58
> > | hopper stand		|	3.58 $\pm$ 1.09 |	4.97 $\pm$ 1.81
> > | reacher easy		|	108.99 $\pm$ 45.01	|	104.76 $\pm$ 28.24

---

### Decision · Program_Chairs · 2021-09-27

**Decision:**

Accept (Poster)

**Comment:**

This paper argues, in RL applications --specifically focusing on SAC algorithm-- typical deeper and wider architectures don't provide big gains that we see in supervised learning. Then paper shows that spectral normalization could be used to improve the performance of deep architectures. The paper focuses on the continuous control problems.

The authors did a very good job during the rebuttal period and they managed to address the most concerns raised by the reviewers. Specifically, the authors have addressed the limited scope of just discussing on SAC by adding DDPG results too.. The paper could be improved by having the idea tested on a wider class of algorithms and with more smoothness controlling techniques, which the authors have identified as the limitations in the work. The results provided in this paper would still be valuable for the NeurIPS community. I would recommend the authors to cite and discuss about the concurrent work suggested by the reviewer FNPT [1].

[1] Spectral Normalisation for Deep Reinforcement Learning: An Optimisation Perspective Florin Gogianu, Tudor Berariu, Mihaela Rosca, Claudia Clopath, Lucian Busoniu, Razvan Pascanu, ICML 2021